# Preparation of Hyflon AD/Polypropylene Blend Membrane for Artificial Lung

**DOI:** 10.3390/membranes13070665

**Published:** 2023-07-14

**Authors:** Jie Li, Ting He, Hongyu Chen, Yangming Cheng, Enrico Drioli, Zhaohui Wang, Zhaoliang Cui

**Affiliations:** 1State Key Laboratory of Materials-Oriented Chemical Engineering, College of Chemical Engineering, Nanjing Tech University, Nanjing 210009, China; 202061104016@njtech.edu.cn (J.L.); tinglei@njtech.edu.cn (T.H.); 202161204227@njut.edu.cn (H.C.); 2National Engineering Research Center for Special Separation Membrane, Nanjing Tech University, Nanjing 210009, China; 3Jiangsu Aikemo High-Technology Co., Ltd., Suzhou 215000, China; chengyangmi@163.com; 4Research Institute on Membrane Technology, ITM-CNR, Via Pietro Bucci 17/C, 87036 Rende, Italy; e.drioli@itm.cnr.it

**Keywords:** polypropylene, thermally induced phase separation, environmentally friendly diluent, fluoropolymer, ECMO

## Abstract

**Highlights:**

Preparation of anti-wetting polypropylene membrane using environmentally friendly diluents via TIPS methodOptimization of Preparation Process for Polypropylene Hollow Fiber MembraneFluorinated polymer modified membrane enhances anti-wetting performance and blood compatibility

**Abstract:**

A high-performance polypropylene hollow fiber membrane (PP-HFM) was prepared by using a binary environmentally friendly solvent of polypropylene as the raw material, adopting the thermally induced phase separation (TIPS) method, and adjusting the raw material ratio. The binary diluents were soybean oil (SO) and acetyl tributyl citrate (ATBC). The suitable SO/ATBC ratio of 7/3 was based on the size change of the L-L phase separation region in PP-SO/ATBC thermodynamic phase diagram. Through the characterization and comparison of the basic performance of PP-HFMs, it was found that with the increase of the diluent content in the raw materials, the micropores of outer surface of the PP-HFM became larger, and the cross section showed a sponge-like pore structure. The fluoropolymer, Hyflon ADx, was deposited on the outer surface of the hollow fiber membrane using a physical modification method of solution dipping. After modification, the surface pore size of the Hyflon AD40L modified membranes decreased; the contact angle increased to around 107°; the surface energy decreased to 17 mN·m^−1^; and the surface roughness decreased to 17 nm. Hyflon AD40L/PP-HFMs also had more water resistance properties from the variation of wetting curve. For biocompatibility of the membrane, the adsorption capacity of the modified PP membrane for albumin decreased from approximately 1.2 mg·cm^−2^ to 1.0 mg·cm^−2^, and the adsorption of platelets decreased under fluorescence microscopy. The decrease in blood cells and protein adsorption in the blood prolonged the clotting time. In addition, the hemolysis rate of modified PP membrane was reduced to within the standard of 5%, and the cell survival rate of its precipitate was above 100%, which also indicated the excellent biocompatibility of fluoropolymer modified membrane. The improvement of hydrophobicity and blood compatibility makes Hyflon AD/PP-HFMs have the potential for application in membrane oxygenators.

## 1. Introduction

After the outbreak of Corona Virus Disease 2019 (COVID-19), people gradually realized the significance of Extracorporeal Membrane Oxygenation (ECMO [1]) in saving the lives of critically ill patients. ECMO named extracorporeal membrane oxygenation, also known as artificial lung, temporarily replaces cardiopulmonary function to gain valuable time for the treatment of patients with severe cardiopulmonary disease [2]. Compared with the gel hydrophilic material of artificial blood vessel [3] and the nano-organic–inorganic composite of artificial bone [4], the materials used for artificial lungs are mainly hydrophobic materials of polyolefins for long-term blood oxygenation especially in commercial products.

Polypropylene (PP) membrane is one of the most commonly used microporous membranes. As having certain hydrophobicity and biocompatibility, PP membranes are not only widely used in water treatment, gas separation, and membrane crystallization but also are used in biomedical fields such as hemodialysis, blood separation, biosensors, and blood oxygen exchangers, especially membrane oxygenators [5]. Although polypropylene air–blood exchange membranes have a short service life or may leak blood during use [6], they are still the main oxygenator material in short-term extracorporeal life support (ECLS) applications, such as the treatment of neonatal pulmonary disease [7] and cardiac surgery [8]. At present, the main artificial lung membranes on the market are PMP and PP materials [9]. The raw material price of PMP is more than ten times that of PP, and currently only Mitsui Chemical Company in Japan sells it globally, while PP raw materials have many medical grade brands [10]. Therefore, many commercial ECMOs still use polypropylene microporous membrane oxygenators, such as the Affinity Nt of Medtronic [11], the Quadrox-i Adult of McVeigh [12], and the A.L.One AF PLUS of Eurosets [13].

As a well-established preparation method, thermally induced phase separation (TIPS) is usually used to prepare polypropylene microporous membranes, which have more uniform pores and fewer defects than cold-melt or hot-melt or both stretching methods [14]. However, in previous preparation experience, we found that ibutyl phthalate (DBP) and diphenyl carbonate (DPC) and other phthalate-containing solvents that are toxic to the environment and human body are still in use. In recent years, there have been a large number of reports on the use of green diluents for membrane preparation [15,16,17], as shown in Table 1.

A PP membrane has certain biocompatibility, but thrombosis and hemolysis will inevitably occur when the membrane is in contact with blood. Therefore, a large number of studies have been conducted to modify the membrane in different ways to improve the blood compatibility of the membrane [5]. In addition, commercial membranes such as the Balance Biosurface of Medtronic Affinity Fusion Oxygenator [23] add special biological coatings on the membranes.

Fluoropolymers are a class of polymers with excellent hydrophobicity and oleophobicity exhibiting good biocompatibility [24,25]. Fluoropolymers that have been used in medical devices include PTFE [26], FEP [27], PFA [28], etc. The reasons for the good biocompatibility of fluoropolymers may be the following two points: (1) The low surface energy of fluoropolymers reduces the adhesion of the material surface to proteins and blood cells, thereby prolonging clotting time and reducing thrombosis [29]. (2) The zeta (*δ*) potential value on the surface of endothelial cells in normal human blood vessel walls is from −8 to −13 mV [30], and blood cells typically have a negative charge, while fluoropolymers have a strong negative charge. The surface of materials attached to fluoropolymers will repel blood cells and not easily adhere, thereby reducing the formation of thrombi.

Hyflon ADx (Figure 1 shows the chemical structure formula), as a member of fluoropolymers, also possesses the above characteristics. Hyflon ADx is used to prepare composite/blend membranes such as PVDF and PFPE due to its high hydrophobicity. The stable and efficient performance of the composite/blend membrane can achieve Zero Liquid Discharge (ZLD) in direct contact membrane distillation (DCMD) [31] and vacuum membrane distillation (VMD) [32,33]. Ahrumi Park et al. applied Hyflon AD60X to the surface of PVDF co HFP membrane for the first time, significantly improving the liquid entry pressure and wetting resistance of the pristine membrane, while reducing protein adsorption [34]. There are currently no researchers exploring the blood related applications of fluoropolymer coating on PP microporous membranes, so this article aimed to fill this gap.

In this work, we used the environmentally friendly solvent ATBC to prepare polypropylene hollow fiber pristine membranes for an artificial lung and explore the phase separation mechanism of SO and ATBC binary diluent controlling properly the membrane structure and pore size. On the basis of the prepared pristine membrane with good performance, hydrophobic modification was carried out using fluoropolymers, and suitable coating conditions for Hyflon ADx were researched to prepare a polypropylene hollow fiber blend membrane with high hydrophobicity, low surface energy, and a smooth surface. The blend membrane was tested for protein adsorption, platelet adsorption, coagulation time, and hemolysis rate to preliminarily determine its biocompatibility. However, further blood compatibility assessments and animal oxygenation experiments are required for formal clinical use.

## 2. Experimental

### 2.1. Materials

Polypropylene (iPP) (MFR = 3~8 g/10 min) was provided by LyondellBasell Industries (Rotterdam, The Netherlands). Soybean oil (SO) was purchased from Macklin Biochemical (Shanghai, China). Acetyl tributyl citrate (ATBC) was purchased from Yousuo Chemical Technology (Shandong, China). Ethanol and n-pentane were purchased from Sinopharm Chemical Reagent (Nanjing, China). α-Bromonaphthalene was purchased from Aladdin Reagent (Shanghai, China). Hyflon AD40H, Hyflon AD40L, and Hyflon AD60 (Viscosity of them is 120 ± 30, 40 ± 10, and 40 ± 10 cc·g^−1^, respectively) were supplied by Solvay Specialty Polymer (Bollate, Italy). BSA power was obtained from Aladdin (Nanjing, China). Phosphate buffered saline (PBS) buffer solution (pH = 7.4) was purchased from Solarbio (Beijing, China). BCA Protein Assay Kit was provided by cwbio (Nanjing, China). Calcein AM (1000×) was supplied by Beyotime Biotechnology (Shanghai, China). Self-made platelet diluent, fresh venous blood of mice, human umbilical vein endothelial cell, and CCK-8 test solution were obtained from Nanjing Medical University.

### 2.2. Determination of Phase Diagram

In the thermodynamic phase diagram of PP-binary diluent, the double knot line was determined by the cloud point temperature, which was measured by the hot stage polarizing microscope (XPV-800E, Bimu Instrument Co., Ltd., Tianjin, China). A cooling sample of the casting solution, with the size of fine sand, was placed between two cover slides on the 190 °C hot table. At this point, the sample was observed by adjusting the focal length under the microscope eyepiece. Then, the sample was cooled from 190 °C to 90 °C at a cooling rate of 10 °C/min. The cloud point temperature was the temperature at which the casting solution in the microscope field of vision changes from transparent to turbid.

The crystallization temperature was determined by differential scanning calorimetry (TA Q-20, Newcastle, DE, USA). The casting solution cooling sample was placed at 190 °C for 5 min and then cooled to 50 °C at a cooling rate of 10 °C/min. The measured peak value of DSC curve was the crystallization temperature at this time.

### 2.3. Preparation and Modification of PP Fiber Hollow Membrane

#### 2.3.1. Preparation of PP Pristine Membranes

The preparation process of PP-HFMs via TIPS is shown in Figure 2a. Firstly, PP raw material and mixed diluent (SO/ATBC) were stirred evenly by the agitator and added to the raw material inlet of the screw extruder. Next, the raw material and diluents were added to form the substrate precursor through a heating section of 160~190 °C in the heating part of the screw extruder. Then, the hollow fiber-like matrix precursor was solidified and formed in a 40 °C cooling bath. Subsequently, the membrane fibers were subjected to alternating hot and cold stretching in a rinse bath at 60–80 °C. After collecting in the collection tank, the membrane fibers were, respectively, immersed in appropriate amounts of n-pentane and ethanol to alternately extract the diluent in the membrane. Finally, the final PP-HFM product was obtained by drying.

#### 2.3.2. Hyflon ADx Modification

Hyflon ADx polymer solutions with different mass concentrations were prepared at 60 °C using 3M HFE-7100 and Hyflon ADx. Several PP-HFMs with lengths of 5–10 cm and sealing at both ends were selected and immersed in a cooling Hyflon ADx solution for at least 20 min. Finally, the membranes were dried in an oven for 4 h after taking out from the fluoropolymer solution. The multiple coating modifications were repeated the above steps. The experimental process was shown in Figure 2b.

### 2.4. Basic Characterization of PP Membranes

#### 2.4.1. Morphologies

The morphologies of PP hollow fiber membranes were examined using a scanning electron microscope (SEM, S-4800, HITACHI, Tokyo, Japan). PP membranes were fractured in liquid nitrogen before being sputter coated with an ultrathin gold layer. The SEM with the accelerating voltage of 3.0 kV was used to examine the cross-section and surface morphologies of membranes.

#### 2.4.2. Membrane Mechanical Properties, Gas Permeability, and Pore Size Distributions

Mechanical properties of PP membrane

Mechanical strength tests were performed with a tensile strength testing instrument (Model SH-20, Dushan Instrument Co., Ltd., Zhejiang, China). Then, tensile strength (*σ*) and elongation at break (*δ*) were calculated by Equations (1) and (2):(1)$$\sigma =\frac{F}{A_{cross\;section}}$$
(2)$$\delta =\frac{L-L_{0}}{L_{0}}\times 100\%$$
where *A_cross section_* (mm^2^) is the cross-sectional area of the membrane sample, *L* (cm) and *F* (N) are the final length and tensile stress when the sample is broken, and *L*_0_ (cm) is the initial length of the tested sample.

Gas permeability

A PP-HFM 3–5 cm-long was fixed in the self-made measuring component as shown in Figure 3. At room temperature and with 0.1 MPa (1 bar) gas pressure applied, the permeation flux of membrane O_2_ and CO_2_ was measured using a soap bubble flowmeter. The gas flux was calculated as Equation (3):(3)$$\left(\frac{P}{l}\right)=\frac{Q}{A\Delta P}$$
where *P* (bar) is the measuring pressure, *l* (cm) is the length of the PP membrane, $\frac{P}{l}$ is the gas flux of membrane (mL·cm^−2^·min^−1^·bar^−1^, 1 mL·cm^−2^·min^−1^·bar^−1^ ≈ 222.22 GPU), *Q* is the gas flow rate through membrane per unit time, *A* (cm^2^) is the apparent surface area of membrane, and ∆*P* (bar) is the pressure difference on both sides of membrane.

Pore size distribution and porosity

The pore size distribution can be determined using the mercury intrusion porosimetry (MIP) method. This method is a type of bubble point method, which is based on the principle that mercury does not wet general solids. To make the pores wet, external pressure needed to be applied. The larger the external pressure, the more mercury can enter the pores with smaller pore radii. The volume of the corresponding pore size can be determined by measuring the amount of mercury entering the pore under different external pressures. We used the Mercury Porosimetry Analyzer (PoremasterGT60, Quantum Instrument, Boynton Beach, FL, USA) to determine the pore size distribution of PP membranes. The software Porewin (vision 5.0, Quantum Instrument, Boynton Beach, FL, USA) was used to process the obtained data.

The porosity of PP membrane was determined by dry–wet membrane weight method. The measurement principle is to measure the ratio of the membrane pore volume to the entire membrane volume according to the weight of the membrane before and after soaking in solvent. First, the membrane was cut into a specific shape, and then the dry membrane was weighed recording its weight as *W_d_*. Next, the dry membrane was soaked in kerosene for 24 h. Before recording the weight of the wet membranes as *W_w_*, kerosene was wiped off from the surface. Finally, the porosity (*ε*) of the membrane can be calculated by Equation (4):(4)ε=WW−WdρkWW−Wdρk+WdρP×100%
where *ε* is the porosity of PP membrane (%); *W_w_* is the mass of the wet membrane after wetting (g); *W_d_* is the mass of the dry membrane (g); *ρ_P_* is the density of PP (0.9 g·cm^−3^); *ρ_K_* is the density of kerosene (0.8 g·cm^−3^).

The pore size distribution can also be determined using the Brunauer–Emmett–Teller (BET) method. By measuring the isothermal adsorption curve of gas (N_2_), the pore size distribution, total pore volume, and average pore size of the membrane can be sequentially calculated by the Kelvin equation, which corresponds to the pore size and adsorbate pressure that produces adsorption condensation or depolymerization. The multi-station physical adsorption instrument (Micromeritics ASAP2460, Norcross, GA, USA) was used to measure the changes in membrane pore size before and after modification of PP membranes.

#### 2.4.3. Contact Angle and Surface Energy

(1)Contact angle

A PP-HFM was fixed onto a glass slide and placed into a contact angle (CA) tester (Dataphysios OCA25, Haishu Maishi Testing Technology Co., Ltd., Ningbo, China). The contact angles of PP-HFMs were then measured with pure water and α-bromonaphthalene.

(2)Surface energy

The surface energy of polymer materials is usually measured using the Owens-Wendt (OW) method, which is more accurate for non-polar or low-polar polymers [35]. Specifically, by measuring polar liquids (Pure water) and non-polar liquids (α-bromonaphthalene) on the surface of the membrane, the surface free energy and components of the test liquid are shown in Table 2. The contact angle of bromonaphthalene was calculated using the following Equations (5) and (6):(5)$$\gamma_{L}\left(1+cos\theta \right)=2{\left(\gamma_{S}^{d}\gamma_{L}^{d}\right)}^{\frac{1}{2}}+2{\left(\gamma_{S}^{P}\gamma_{L}^{P}\right)}^{1/2}$$
(6)$$\gamma_{S}=\gamma_{S}^{d}+\gamma_{S}^{P}$$
where *γ_S_*, *γ_L_* represents the free energy of the solid surface and the free energy of the liquid surface, respectively, *γ_S_^d^*, *γ_S_^p^* is the solid dispersion force and polar force terms, and *γ_L_^d^*, *γ_L_^p^* is the dispersion force and polarity force terms of the liquid. PP-HFMs’ surface energy *γ_S_* were obtained by using the simultaneous formula.

(3)Average roughness

#### 2.4.4. Membrane Wetting

The membrane wetting test of membrane module and device diagram was shown in Figure 4. The membrane module was obtained by fixing 5 PP-HFMs in a PVC pipe with polyurethane adhesive (Figure 4a). When starting the measurement, we controlled the air purge pressure at the inlet end of the membrane module to 0.01 MPa and the pure water pressure outside the PP-HFMs at 0.1 MPa. The initial air humidity inside the PP-HFMs was adjusted to about 50%, and then the device was allowed to run for about 40 min, and the change of air humidity inside the membrane was recorded with a humidity recorder (TP500V4.0, Toprie Electronics Co., Ltd., Shanghai, China) during the measurement showing in Figure 4b,c).

### 2.5. Biocompatibility Characterization of PP Membranes

#### 2.5.1. BSA Protein Static Adsorption

The protein adhesion of PP pristine membrane and modified membrane was tested through static protein adsorption using BSA as albumin source. Firstly, a membrane sample was cut to 1 cm-long and sealed at both ends (the effective area is approximately 0.13 cm^2^) and then immersed in 2 mL of 0.5 g·L^−1^ BSA-PBS buffer solution incubating at 37 °C for 2 h. Next, the soaking solution of 200 μL BCA working solution was prepared with 1/8 volume of soaking solution. Then, the soaking solution was placed in a constant temperature shaker at 37 °C for 30 min. The concentration of BSA staining solution was determined at 562 nm by using a microplate reader (Synergy H1, Bio-Tek Instruments, Inc., Winooski, VT, USA). In addition, the protein adsorption capacity (MBSA) was calculated according to Equation (7):(7)$$M_{BSA}=\frac{\left(c_{o}-c_{t}\right)V}{A}$$
where *c*_𝑜_ and *c_t_* are the concentrations of BSA before and after the membrane sample immersion, respectively. 𝑉 (𝐿) is the volume of the BSA solution, and *A* (cm^2^) is the effective area of the membrane.

#### 2.5.2. Platelet Adsorption

Platelet dilution was obtained by centrifuging fresh mice whole blood at 1500 r/min for 15 min. The PP-HFMs with both ends sealed and a length of 1 cm were immersed into 100 μL incubating in wet air at 37 °C for 1 h. After washing 3 times with PBS solution, platelet on membranes was fixed by 2.5 wt% glutaraldehyde solution for 1 h. Membrane samples were washed by PBS 3 times and then stained with Calcein AM in the dark room for 15 min. Finally, the membrane sample-stained platelets were washed with PBS 3 times. The adhesion situation of platelets on the surface of membrane sample was observed under the fluorescence microscope (Leica DM6M, Leica AG, Wetzlar, Germany).

#### 2.5.3. Hemolysis Ratio

Blood dilution was prepared by mixing mice whole blood and physiological saline in a 4:5 ratio. PP membranes were immersed in 200 μL diluted blood added with 10 mL of physiological saline and were shaken in a shaking incubator (MQL-61R, Shanghai Minquan Instrument, Shanghai, China) at 37 °C for 1 h. Subsequently, the diluent samples were centrifuged in a centrifuge (SN-LSC-40, Sunne Instrument Equipment Co., Ltd., Zhejiang, China) at a speed of 1000 r/min for 5 min. The supernatant of centrifugal samples was measured the absorbance by using an enzyme-linked immunosorbent assay at 540 nm. Using diluted blood + physiological saline as the negative control and diluted blood + distilled water as the positive control, the hemolysis rate was calculated according to Equation (8):(8)$$Hemolysis\;rate=\frac{A_{Sample}-A_{Positive\;control}}{A_{Negative\;control}-A_{Positive\;control}}$$
where *A_Sample_*, *A_Positive control_*, and *A_Negative control_* (L·g^−1^·cm^−1^) are the absorbances of the dilution sample, the positive control group sample, and the negative control group sample, respectively.

#### 2.5.4. Blood Clotting Time

A total of 100 μL of fresh mouse venous blood drops was added to the surface of a 2 cm-wide PP-HFM pasted side by side on a glass slide. We used the syringe needle to repeatedly pick up the blood droplets until they were clearly drawn and form a blood clot. The clotting time was calculated as the CT, and the average CT value was obtained by repeating 3 times.

#### 2.5.5. Cytotoxicity Testing

Biomedical materials require cytotoxicity testing, which is one of the mandatory testing indicators for ECMO and other clinical applications. The CCK-8 (Cell Counting Kit-8) [37] and MTT assay [38] are both used to measure the number of live cells in cell proliferation or toxicity experiments, but the former has advantages such as being more sensitive, being non-toxic to cells, having simple steps, and having stable performance. Therefore, in this article, we selected the CCK-8 assay to detect cell activity. The PP pristine membrane and Hyflon AD40L modified membrane were added to the human umbilical vein endothelial cell culture medium for 24 h, and then the light absorption value at 450 nm was measured with the microplate reader. The cell survival rate was calculated with the blank group as the control.

We measured the absorbance value of the cell culture sample at 450 nm and calculated its cell survival rate using the following Equation (9):(9)$$Cell\;survival\;rate=\frac{OD_{Sample}-OD_{Blank}}{OD_{Control}-OD_{Blank}}$$
where *OD_Sample_* is the test value of experimental material sample, *OD_Blank_* is the blank sample, and *OD_Control_* is the test value of control sample. *OD_Sample_* − *OD_Blank_* is absorbance of the PP pristine membrane and the Hyflon AD40L/PP-HFM, and *OD_Control_* − *OD_Blank_* is absorbance of blank control group.

## 3. Results and Discussion

### 3.1. Phase Diagram

The interaction between SO and iPP is weak, but the casting solution composed of them will pass through a large S-L (solid–liquid) phase separation area when forming the membrane, which leads to the appearance of a spherical structure inside the PP membrane and exhibits poor mechanical performance due to hardness and brittleness [39]. After adding ATBC to the diluent, the spherical structure within the membrane was reduced to some extent. As shown in Figure 5, when the PP concentration was fixed at 25 wt% and 30 wt%, the change trend of cloud point temperature in the PP-SO/ATBC thermodynamic phase diagram was the same. The difference between cloud point temperature and crystallization temperature in the phase diagram increased from about 19 to 40 and then decreased to about 10, indicating that the interaction between PP and the whole diluent was first weakened and then strengthened. The cloud point temperature formed a double nodal line, while the crystallization temperature constituted the crystallization curve. The area between the two temperatures represented the L-L desired separation area. It can be seen that when the ratio of SO/ATBC was around 7 to 3, the casting solution would obtain a larger L-L phase separation area during the membrane forming process, thereby obtaining more continuous pores.

### 3.2. Comparison of Self-Made PP Membranes

According to the thermodynamic phase diagram analysis in Figure 5, it can be concluded that in order to obtain a larger liquid–liquid separation region, the SO/ATBC diluent ratio was around 7/3. Therefore, we used this ratio of diluent solution, adjusted the ratio between polymer/diluent, and the parameters related to the preparation of hollow fiber membranes to obtain three different PP-HFMs of M1-3. Some basic characteristics of self-made and commercial PP membranes are shown in Table 3.

From the characterization results of PP membrane performance in Table 3, we can see that with the increase of the content of membrane preparing diluent, the tensile strength of PP membrane decreased, but all of them are above 10 MPa. The tensile rate decreased by 1800% from about 2200%. The gradual increase in porosity and gas permeation flux of M2 and M3 membranes indicated an increase in membrane pore size, which can also be seen from the changes in membrane surface pore size in Figure 6.

Compared to commercial PP membranes, the self-made membranes (M2 and M3) had the better performance except for slightly lower tensile strength from the above characterization analysis. From the SEM images of PP membranes’ structure morphology in Figure 6, self-made membranes had a sponge like structure with smaller surface pores and a uniformly distributed cross-section compared to the commercial membrane.

The pore size distribution curves of self-made membranes M2 and M3 and commercial PP membranes are shown in Figure 7. M2 has a relatively concentrated pore size distribution of about 140 nm. Due to the large pore size near the outer surface of membrane, M3 has some pore sizes concentrated around 330 nm, in addition to the internal pores of about 80 nm. The commercial membrane is similar to M3, with even larger surface pores ranging from 3 to 4 μm and defects.

Both self-made membranes M2 and M3 can be used as hydrophobic modified pristine membranes, but the larger surface pores mean that the membrane surface is more easily penetrated and wetted by water. Therefore, we used a self-made PP membrane M2 with slightly smaller surface pores and moderate permeation flux for subsequent hydrophobic modification.

### 3.3. Determination of PP Membrane Modification Conditions

To determine the most suitable Hyflon ADx hydrophobic modification conditions for the membrane, the appropriate coating concentration and number of coating times were first determined using Hyflon AD40H. Then, the hydrophobicity of different types of Hyflon ADx-modified membranes under the same coating concentration and number of coating times was compared to determine the modification conditions.

#### 3.3.1. Coating Concentration and Coating Frequency of Hyflon ADx

From the surface morphology of the SEM modified membrane in Figure 8, the pores on the membrane surface decreased or even disappeared with the increase of coating concentration and coating time. Due to the low coating concentration, from the two cross-sectional SEM images on the right, there was no obvious dense layer on the cross-section near the surface of Hyflon AD40H-modified membrane compared to the pristine membrane.

During the process of gas passing through a membrane containing fluoropolymer, the inlet pressure is positively correlated with the flow rate, which is to say that the pressure is positively correlated with the flux [40]. Therefore, we chose to measure the permeation flux of the membrane at the same initial pressure (0.1 MPa). From gas permeation flux changes of the Hyflon AD40H-modified membrane shown in Table 4, it can be seen that the permeation flux of the 1#-0.05 wt% and 2#-0.02 wt% modified membranes is equivalent, and both were more stable than the membranes modified with low coating concentration and low coating times. As the coating concentration and coating time further increased, the gas flux of the modified membranes under these two modified conditions approached 0.

The hydrophobicity of the membrane surface does not change significantly when single coating with different concentrations of Hyflon ADx. As shown in Figure 9a, the contact angle of the PPmembrane was increased by about 8° after a single coating of 0.10 wt% Hyflon AD40H, while it was increased by about 14° during the same concentration of secondary coating. The Hyflon AD40H modified membrane also exhibits significant contact angle values under two modified conditions (1#-0.05 wt% and 2#-0.02 wt%) with comparable gas permeation flux.

Subsequently, we conducted hydrophobicity tests under these two modified conditions to further compare their hydrophobicity. From Figure 9b, it can be seen that if the air humidity inside the membrane increased to 70%, the commercial PP membrane can be tested for about 1 min, while the self-made PP membrane took about 3 min, indicating the advantage of TIPS method compared to melt stretching method in which TIPS preparation can obtain PP membrane with smaller surface pores and fewer defects. Similarly, it took about 5 min for the air humidity inside the 1#-0.05 wt% Hyflon AD40H modified membrane to reach 70%, while the 2#-0.02 wt% Hyflon AD40H modified membrane took 15 min. This indicated that multiple coatings at low concentrations can enhance the adhesion of fluoropolymers to PP membranes and also indicated that PP membrane under the modification condition of 2#-0.02 wt% was more difficult to wet. Therefore, the coating concentration and number of coating times of the fluoropolymer were roughly determined.

#### 3.3.2. Modification Types of Hyflon ADx

In Figure 10, under the same coating concentration and times (2#-0.02 wt%), the SEM morphologies of different types of Hyflon ADx modified PP membranes are basically the same. Due to the low concentration of fluoropolymer coating, there was no obvious dense layer on the cross-section, but Hyflon ADx adhered to the membrane surface because of the reduction of surface pores. The decrease in surface pore size and number of pores meant that the modified membrane surface is more difficult to wet.

As the 3D images of different PP membrane surfaces measured by AFM are shown in Figure 11, the protrusion on the surface of Hyflon ADx modified membranes had been reduced, which shows that fluoropolymer smoothed the pristine membrane surface. From the changes in Ra values in Table 5, it can be seen that the average surface roughness of the modified membrane had decreased by about 10–15 nm, with a maximum reduction of 14.69 nm in Ra values for the Hyflon AD40L modified membrane. The coating of fluoropolymer not only compensates for some surface defects of PP pristine membrane but also enhanced the hydrophobicity and reduced roughness and surface energy of pristine membrane.

In summary, 2#-0.02 wt% Hyflon AD40L modified membrane had stronger hydrophobicity compared to the pristine and commercial PP membrane and had lower surface energy and roughness (17.70 mN·m^−1^ and 17.19 nm). So, this modification condition is the most suitable.

#### 3.3.3. Comparison between Pristine Membrane and Hyflon ADx Modified Membrane

(1)Pore size distribution and pore volume distribution

The BET method generally measures micropores with a pore size of 0–50 nm. Although the pore size of the PP membrane we prepared is concentrated between 100 and 200 nm, some conclusions can also be drawn from the changes in pore size distribution and pore volume from the small pore size. In Figure 12, the pore size distribution and pore volume distribution curve of the modified membrane were both located in the following description of pristine membrane. It is true that Hyflon AD40L was coated on the surface of pristine membrane and covered the surface facial mask holes and even some of the internal holes. Figure 12b shows that the pore volume of the two types of PP membranes increases with the increase of pore size, indicating that the pore size is not concentrated within the measured range but at a larger pore size.

(2)XRD analysis

From Figure 13, it can be seen that the peak of XRD spectrum before and after PP membrane modification is α characteristic peaks of crystalline polypropylene (2 θ = 14.0°, 16.7°, and 18.5°). Both high spectral intensities indicated that both membranes had more α mainly crystalline (>10,000). The absence of the characteristic peak of Hyflon AD40L in the spectrum of the modified membrane in the figure was due to the low concentration of fluoropolymer adhering to the membrane surface in an amorphous form. Therefore, the presence of Hyflon ADx cannot be detected by using XRD.

### 3.4. Blood Compatibility of PP Membranes

#### 3.4.1. BSA Protein Static Adsorption and Hemolysis Ratio Assay

Within a few milliseconds after the membrane material contacts with biological fluids such as blood, proteins adhere to the membrane surface in the form of adsorption, and the adsorbed proteins will undergo single displacement reaction with proteins with higher affinity [41]. The adsorbed proteins can mediate platelet cell adhesion and aggregation, resulting in thrombus formation [42,43]. Especially in membrane oxygenators, the gas–blood exchange membrane needs to have certain anti-protein adhesion properties; otherwise, there will be coagulation and thrombosis problems caused by protein adhesion. This not only greatly shortens the service life of membrane oxygenators but also poses a serious threat to critically ill patients.

Albumin is the most important protein in human plasma, which maintains nutrition and osmotic pressure of the body. The concentration reaches 38–48 g/L, accounting for approximately 50% of the total plasma protein [44]. Due to the high cost of human plasma albumin, we used commonly used bovine serum protein (BSA) as a substitute for determining the protein adsorption capacity of membrane materials.

Both the protein adsorption capacities of self-made and commercial PP membranes range from 1.27 to 1.30 mg·cm^−2^ in Table 6. After hydrophobic modification of the base membrane, the protein adsorption capacity decreased to about 1.04 mg·cm^−2^. A hydrophobic membrane surface implies a higher protein adsorption capacity, but a hydrophobic surface with low surface energy does not directly lead to higher protein adsorption [34]. Hyflon ADx modification reduced the surface energy of PP pristine membrane to around 17 mN·m^−1^, while the smoothed surface and reduced average roughness of the membrane both reduced the adsorption capacity of BSA. In addition, the reduction of membrane surface defects also led to a decrease in protein adsorption capacity.

When studying the anticoagulant properties of materials, it is necessary to consider the degree of damage caused by the influence of materials on red blood cells, which is expressed by the hemolysis rate. If the hemolysis rate exceeds the standard, the material may cause unsafe factors of hemolysis when used in vivo or in vitro. From Table 6, we can see that the Hyflon AD40L modified PP membrane can reduce the hemolysis rate to below the standard of 5%, indicating that fluoropolymer coating modification can to some extent alleviate the damage of the membrane to blood.

#### 3.4.2. Platelet Adsorption and Coagulation Time Determination

Platelets contain coagulation factors, and the contact between platelets and surface of foreign substances (membrane) directly triggers a series of coagulation reactions [45]. Therefore, platelet adhesion is one of the tests for evaluating the blood compatibility of materials.

Here, we used platelet diluents prepared from whole blood of mice for measurement. We can see in Figure 14 that the smaller yellow lights spotted on the surface of Hyflon AD40L modified membrane are significantly less than those on self-made pristine membrane surface (the size of platelets is about 2~8 μm, and chartreuse is displayed by Calcein AM [46]), which indicated that fluoropolymers can reduce platelets adhesion of membrane. However, platelets can enter the macropores on the surface of commercial PP membranes, which affected the determination of adsorption status. Therefore, no comparison was made with the two membranes mentioned above.

Clotting time (CT) [25] refers to the time when blood leaves the blood vessel and undergoes coagulation outside the body. CT is mainly used to determine whether various coagulation factors in the endogenous coagulation pathway are lacking, whether their function is normal, or whether there is an increase in anticoagulant substances [47]. According to the source of the sample, clotting time measurement includes capillary blood sampling method and venous blood sampling method [48]. The method we use here is the second one.

We used the venous blood of fresh mice to measure the CT values of different PP membranes. In Figure 15a, Hyflon AD40L modified PP membrane had a longer clotting time (873 s) and exhibited a slightly better effect than commercial membranes in delaying coagulation. Correspondingly, we can also see in Figure 15b that the blood on the surface of modified membranes was elliptical and spherical which was more hydrophobic, while the self-made pristine membrane and commercial membrane were more susceptible to blood infiltration. Coagulation was related to the adhesion of platelets, and less platelet adhesion prolongs the membrane coagulation time. Less platelet adhesion was due to enhanced hemocompatibility of the pristine membrane with Hyflon ADx.

#### 3.4.3. Cytotoxicity Test

As shown in Figure 13, there was no statistically significant difference in absorbance between the Hyflon AD40L/PP-HFM sample and the control group sample, indicating no significant difference in cell activity. From Figure 16, the cell survival rates of the Hyflon AD40L/PP-HFM were about 100.10%, which demonstrated that the adopted fluoropolymer were not cytotoxic and met the requirements for the use of medical materials. The results of cytotoxicity tests have further enhanced the clinical application of fluoropolymer modified membranes.

Although the modification performance of Hyflon ADx effectively improves the blood compatibility and hydrophobicity of PP pristine membrane, the long-term stability of fluoropolymers still needs to be considered for their coating stability. In addition, further blood compatibility evaluation and oxygenation performance testing in vivo are also needed for clinical use, which are our future research work.

## 4. Conclusions

Polypropylene hollow fiber membranes with sponge-like structure, uniform and controllable pore size, small micropores, and fewer surface defects were prepared by thermally induced phase separation method and green binary diluents. The suitable conditions for Hyflon ADx coating modification were determined through experiments such as permeation flux measurement, SEM, contact angle, and the membrane-wetting test. After modification with fluoropolymer Hyflon AD40L, the surface energy of PP pristine membrane was reduced to about 17 mN·m^−1^, and the average roughness was reduced to about 17 nm. At the same time, the decrease of membrane surface pore size and the increase of contact angle both indicated that the hydrophobicity of the modified membrane was improved. Through preliminary blood compatibility evaluation, it was found that fluoropolymer coatings can effectively reduce the adsorption of albumin and platelets on PP membranes and reduce the hemolysis rate to within the standard of 5%, extending the coagulation time. In addition, the 100% survival rate of modified membrane cells in the cytotoxicity test further indicates that the modified membrane has good biocompatibility. Although the hydrophobicity of PP membrane is more important than biocompatibility, it is still necessary to modify the blood compatibility of PP membrane to avoid severe coagulation and immune reactions. Of course, further evaluation of oxygenation performance and blood compatibility is needed for Hyflon AD40L/PP-HFMs to move towards clinical use.

## Figures and Tables

**Figure 1 membranes-13-00665-f001:**
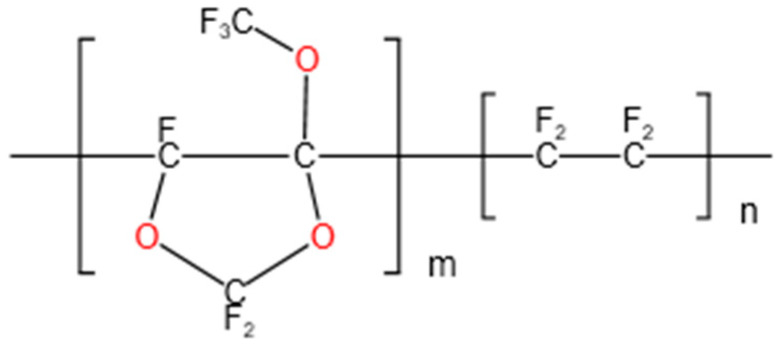
Hyflon AD chemical structure formula: Hyflon AD 40 (m = 0.4, n = 0.6); Hyflon AD60 (m = 0.6, n = 0.4).

**Figure 2 membranes-13-00665-f002:**
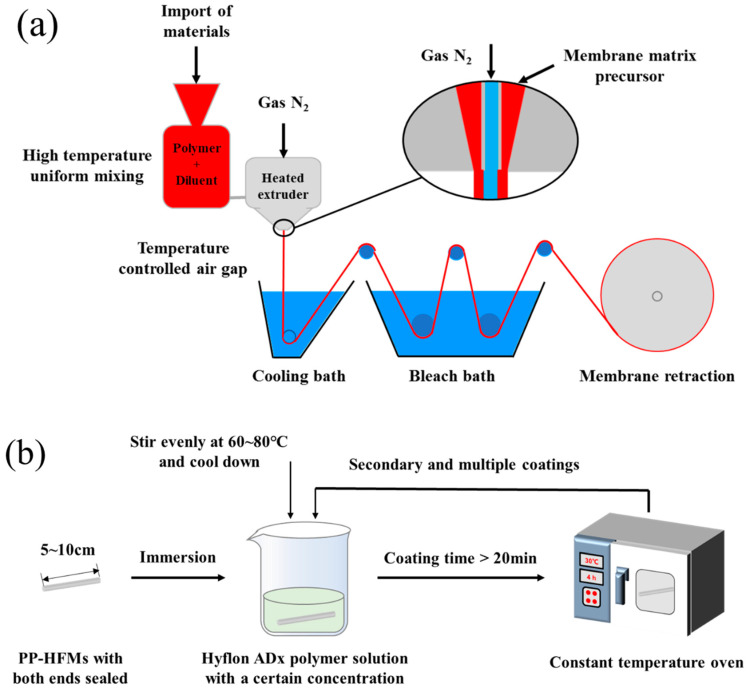
(**a**) Flow chart of PP hollow fiber pristine membranes (PP-HFMs) preparation; (**b**) Flow chart of Hyflon Adx coating modification of PP-HFMs.

**Figure 3 membranes-13-00665-f003:**
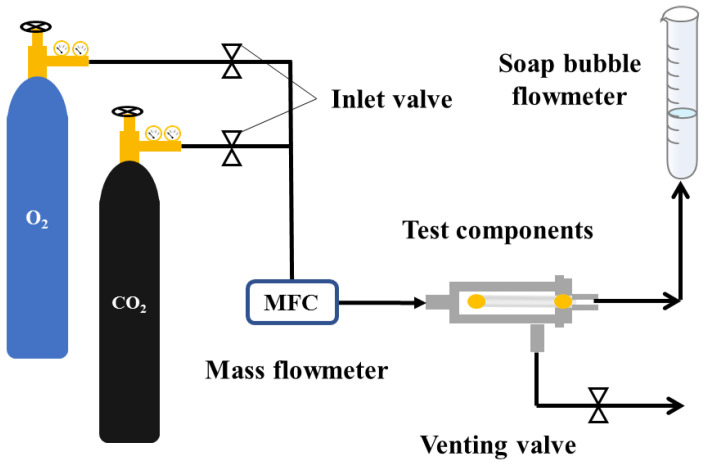
PP hollow fiber membrane gas permeation flux testing device.

**Figure 4 membranes-13-00665-f004:**
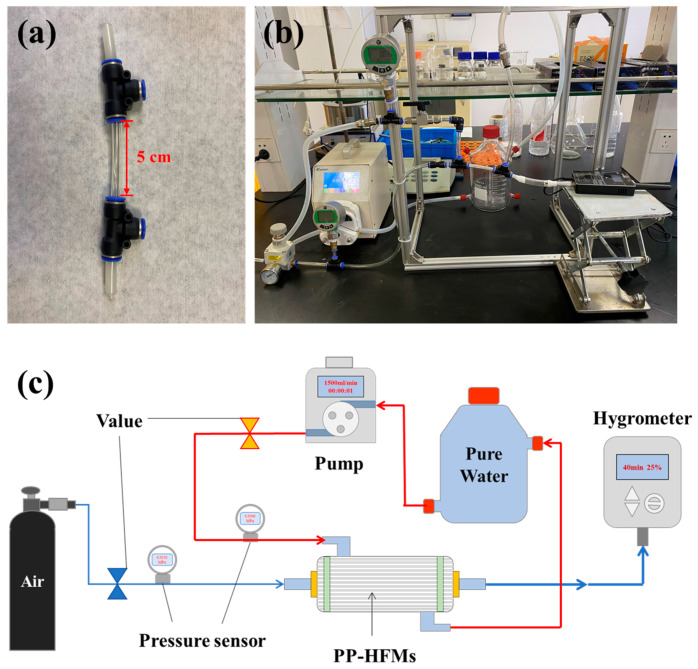
PP membrane wetting test: (**a**) PP-HFM module; (**b**) Physical diagram of the device; (**c**) Schematic diagram.

**Figure 5 membranes-13-00665-f005:**
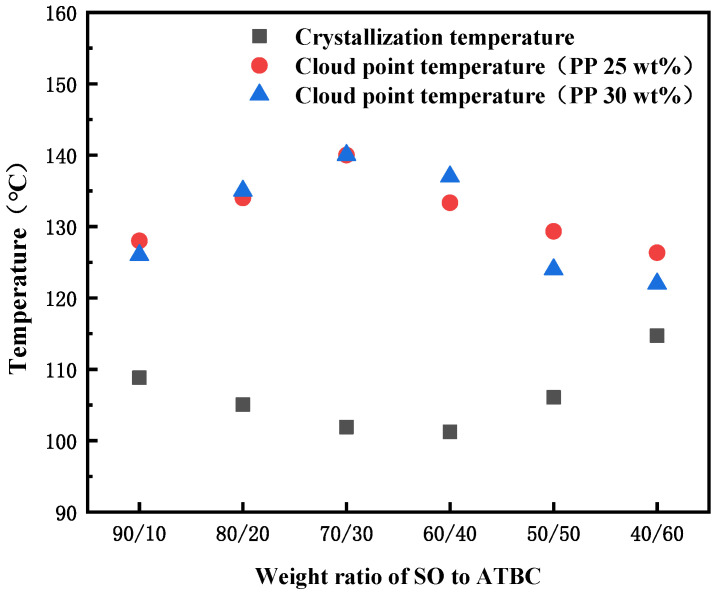
Phase diagram for the PP-binary diluent systems with various weight ratios of SO to ATBC at the PP concentration of 20 and 30 wt%.

**Figure 6 membranes-13-00665-f006:**
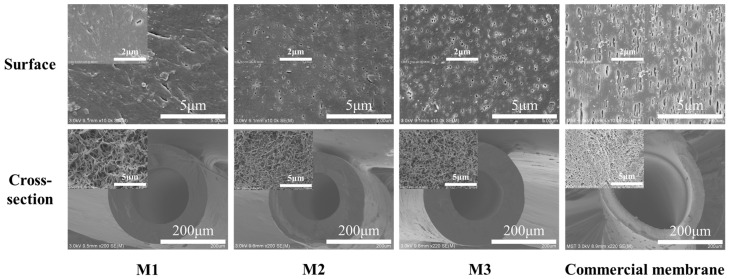
Structure and morphology of self-made and commercial PP membranes.

**Figure 7 membranes-13-00665-f007:**
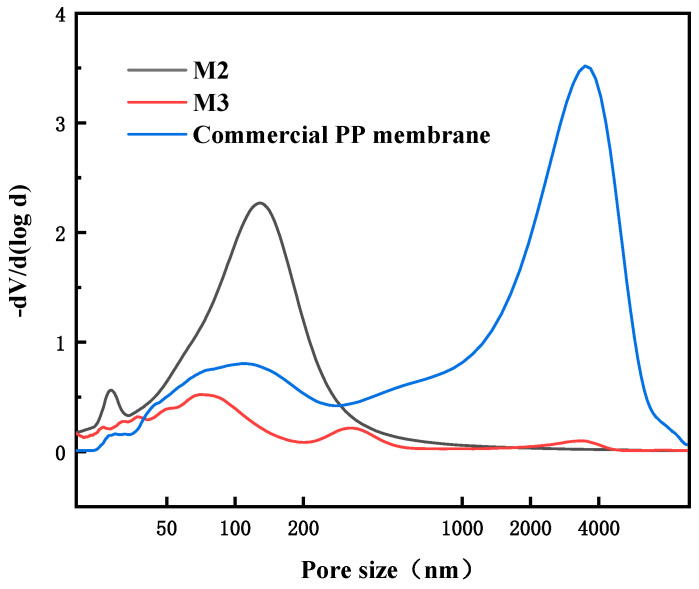
Pore size distributions of different PP membranes.

**Figure 8 membranes-13-00665-f008:**
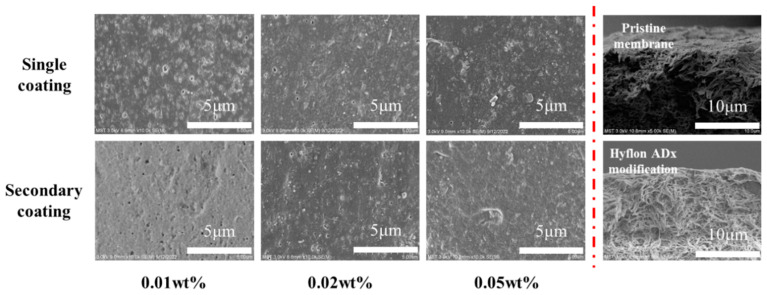
Surface and cross-sectional SEM morphologies of Hyflon ADx-modified membranes with different coating concentrations and coating times.

**Figure 9 membranes-13-00665-f009:**
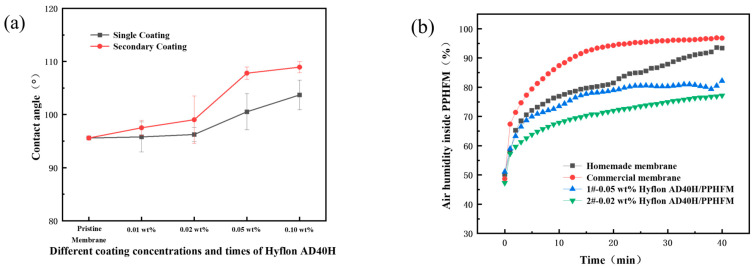
(**a**) Contact angles of membranes modified with different Hyflon AD40H coating concentrations and coating times; (**b**) Change of humidity in different PP membranes with time (1# and 2# represent 1 and 2 coating times, respectively).

**Figure 10 membranes-13-00665-f010:**
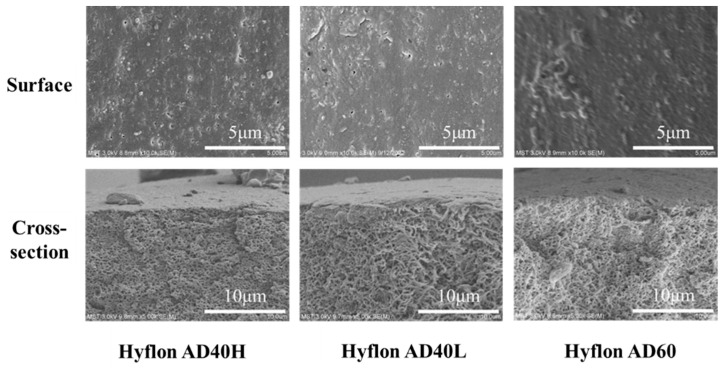
SEM images of PP membranes modified with different types of Hyflon ADx.

**Figure 11 membranes-13-00665-f011:**
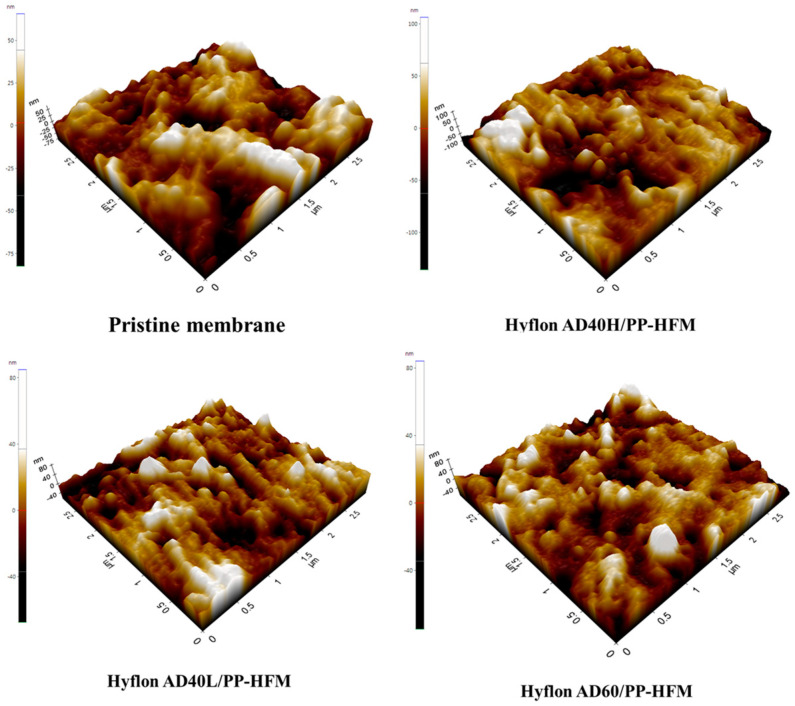
Surface roughness 3D images of PP pristine membrane and different types of Hyflon ADx modified membranes.

**Figure 12 membranes-13-00665-f012:**
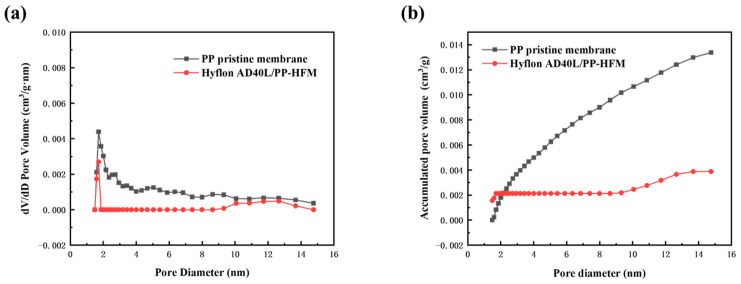
(**a**) Pore size distribution and (**b**) pore volume distribution of PP pristine membrane and Hyflon AD40L modified membrane.

**Figure 13 membranes-13-00665-f013:**
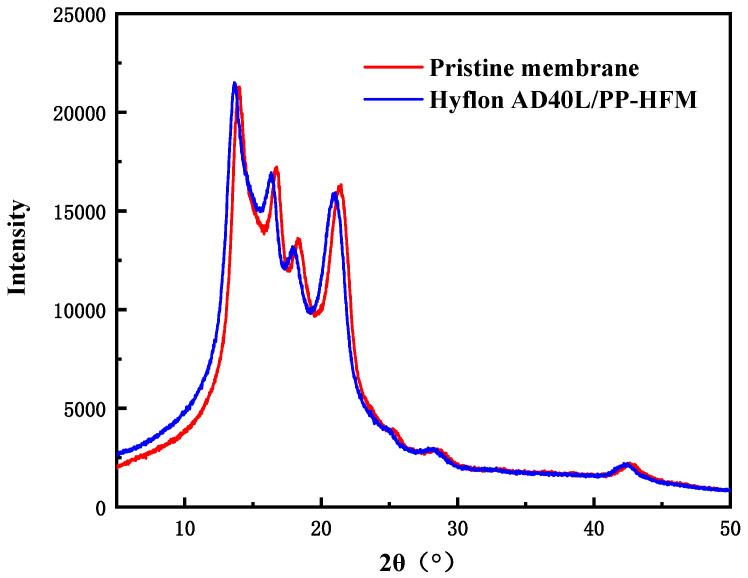
XRD spectra of PP pristine membrane and Hyflon AD40L modified PP membrane.

**Figure 14 membranes-13-00665-f014:**
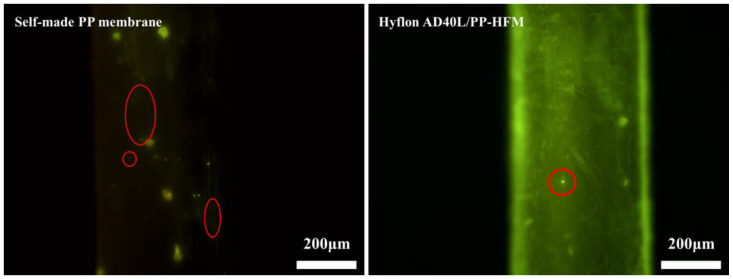
Fluorescence microscopy images of platelet adsorption on self-made pristine membrane and HyflonAD40L modified membrane.

**Figure 15 membranes-13-00665-f015:**
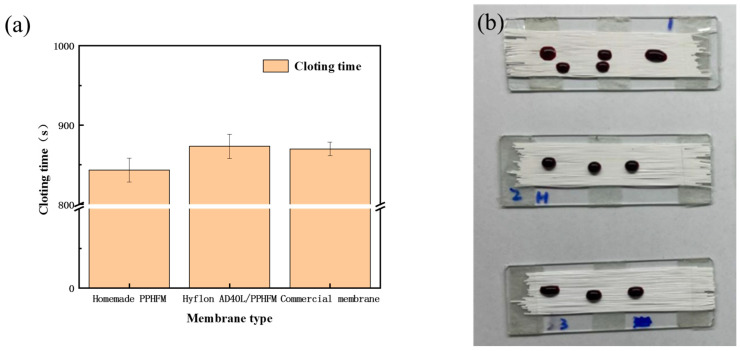
(**a**) Clotting time of different types of PP membranes; (**b**) Coagulation status of different PP membranes (From top to bottom: self-made pristine membrane, Hyflon AD40L modified membrane and commercial PP membrane).

**Figure 16 membranes-13-00665-f016:**
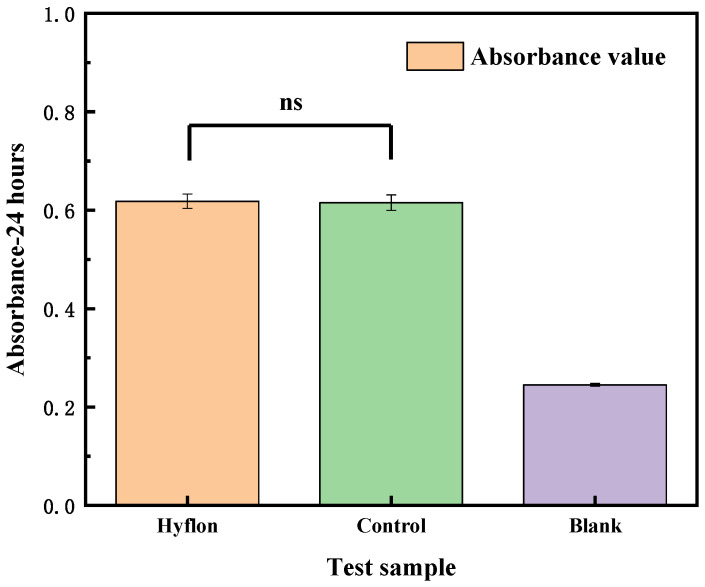
Cytotoxicity test of the Hyflon AD/PP-HFM sample, control group, and blank group sample, respectively.

**Table 1 membranes-13-00665-t001:** Partial summary of diluents used in PP membrane preparation in recent years.

Author	Diluent	Type	Public Year	Ref
Bo Zhou et al.	DPC + Myristic acid	Flat and hollow fiber membrane (HFM)	2015	[18]
Nader Jahanbakhshi et al.	DOP + DBP	HFM	2016	[19]
Juan Wang et al.	DOP + DBP	HFM	2016	[20]
Fang Chuanjie et al.	GTA, DEP, NMP, GBL ect.	HFM	2018	[21]
Yue Sun et al.	DOP + DBP	Flat membrane	2019	[22]

**Table 2 membranes-13-00665-t002:** Surface free energy of test liquid at 20 °C and its $\gamma_{L}^{p}$, $\gamma_{L}^{d}$ components (×10^−5^ N·cm^−1^) [36].

Test Liquid	$\gamma_{L}^{p}$	$\gamma_{L}^{d}$	$\gamma_{L}$	$\gamma_{L}^{p}$ $/\gamma_{L}^{d}$	Non/Polarity
Pure water	51.00	21.80	72.80	2.36	Polarity
α-Bromonaphthalene	0	44.60	44.60	0	Nonpolar

**Table 3 membranes-13-00665-t003:** Performance of PP pristine membranes prepared under different ratios of PP/diluent content.

Membrane Type		M1	M2	M3	Commercial PP Membrane
Ratio of PP/Binary diluent		3/4	2/3	3/7	/
Inner/Outer diameters (mm)		0.19/0.38	0.22/0.38	0.22/0.37	0.28/0.40
Tensile strength (MPa)		12.83	10.98	10.01	16.00
Elongation at break (%)		2257	1876	1893	695
Porosity (%)		45	57	66	45
Gas permeation flux (mL·cm^−2^·bar^−1^·min^−1^)	CO_2_	3.36	10.26	27.57	7.12
	O_2_	3.86	11.68	28.77	8.33

**Table 4 membranes-13-00665-t004:** Gas permeation flux of modified membranes with different Hyflon AD40H coating concentrations and coating times.

Hyflon AD40H Coating Concentration (wt%) and Coating Time (n#)		Gas Permeation Flux (mL·cm^−2^·min^−1^·bar^−1^)	
		CO_2_	O_2_
Single coating (1#)	0	10.78 ± 4.45	11.68 ± 4.10
	0.01	7.53 ± 0.33	8.96 ± 0.78
	0.02	6.88 ± 1.94	7.51 ± 2.56
	0.05	3.59 ± 1.57	3.40 ± 1.26
Secondary coating (2#)	0	/	/
	0.01	3.86 ± 0.83	4.40 ± 1.09
	0.02	3.79 ± 0.55	3.74 ± 0.23
	0.05	/	/

**Table 5 membranes-13-00665-t005:** Contact angle, surface energy, and surface roughness (Ra) of different types of PP membranes.

Membrane Type	Self-Made PP Membrane	Commercial PP Membrane	Hyflon AD40H Modified	Hyflon AD40L Modified	Hyflon AD60 Modified
Pure water contact angle (°)	94.37 ± 2.47	95.77 ± 3.49	96.97 ± 1.15	104.50 ± 3.91	101.70 ± 4.94
Surface energy (mN·m^−1^)	26.91	22.76	21.45	17.70	18.88
Ra (nm)	31.88	/	20.04	17.19	18.30

**Table 6 membranes-13-00665-t006:** BSA protein static adsorption capacity and hemolysis rate of different types of PP membranes (2# indicates a coating time of 2).

Membrane Type	BSA Protein Adsorption Capacity (mg·cm^−2^)	Hemolysis Rate (%)
self-made PP-HFM	1.28 ± 0.27	7.5 ± 0.1
2#-0.02 wt% Hyflon AD40L/PP-HFM	1.04 ± 0.03	4.4 ± 1.3
Commercial PP membrane	1.29 ± 0.48	8.2 ± 0.3

## Data Availability

Not applicable.

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
