# Peer review of "Preparation of Hyflon AD/Polypropylene Blend Membrane for Artificial Lung"

_membranes, 2023, doi:10.3390/membranes13070665_

Round 1

Reviewer 1 Report

Title: Preparation of Hyflon AD/polypropylene composite membrane for artificial lung

In this work, A high-performance polypropylene hollow fiber membrane (PP-HFM) was prepared by using a binary environmentally friendly solvent of polypropylene as the raw material, adopting the thermally induced phase separation (TIPS) method and adjusting the raw material ratio. The binary diluents were soybean oil (SO) and acetyl tributyl citrate (ATBC). The suitable SO/ATBC ratio of 7/3 was based on the size change of the L-L phase separation region in PP-SO/ATBC thermodynamic phase diagram. Through the characterization and comparison of the basic performance of PP-HFMs, it was found that with the increase of the diluent content in the raw materials, the micropores of outer surface of the PP-HFM became larger, and the cross section showed a sponge-like pore struc-ture. The fluoropolymer, Hyflon ADx, was deposited on the outer surface of the hollow fiber mem-brane using a physical modification method of solution dipping.

Well written article. The article has not yet been finalized for acceptance.

I, recommend this paper be published in the Membranes Journal after the authors address the following comments.

·        Review English grammar as there are mistakes throughout the text.

·        Add a section on study limitations before the conclusion section.

·        Numenclature add to the manuscript.

·        The author must improve the introduction with more advanced applications. Also, the author could find new references for the literature review. Such as:

https://doi.org/10.1016/j.actbio.2022.09.003

https://doi.org/10.1016/j.desal.2017.09.030

·        Hyflon Membrane studies need deeper discussion of the results using following relevant references and must be included,

CFD simulation of He/CH4 separation by Hyflon AD60X membrane

Blood oxygenation using fluoropolymer-based artificial lung membranes

·        The physical explanation of figures 9-11 is limited. Please explain more.

·        The conclusion is weak. Please explain your main findings quantitatively.

·        In general, the article does not provide enough evidence for the conclusions presented. Please provide XRD patterns in actual size for further evaluations. How was the peaks intensity? The author should explain the changes in XRD patterns.

·        BET diagrams should be added to the article. Pore size distribution, pore volume distribution and adsorption/desorption results.

·        Membrane roughness is essential. Surface roughness values should be included (AFM graph).

·         The author should compare their results with other membranes. A table or graph would be useful.

·        No details on the reactor design, the real test setup should be presented in the manuscript.

·        How are the changes in membrane surface after the separation process?

·        The authors did not perform the membrane’s regeneration test?

In conclusion, this paper might be made suitable for publication in this Journal if the as-mentioned comments are clarified. These constitute a Major revision of it.

Minor editing of English language required

Reviewer 2 Report

The information presented in the manuscript, Preparation of Hyflon AD/polypropylene composite membrane for artificial lung, is related to the construction of a modified membrane that possesses good blood compatibility and can be used for membrane oxygenators (artificial lung) in contact with blood. This manuscript has contained sufficiently new information, and information is presented in a logical order. Following minor proofreading is needed to further improve the readability of the manuscript.

(a)  Platelet adsorption can be better studied using other biological assays compared to used Calcein AM based protocols. (b) Figure 12 may not clearly demonstrate semi-quantity data of platelet adsorption on self-made pristine membrane and HyflonAD40L modified membrane. (c) Figure 7 can be draw in better way, it is difficult to see inset information and labeling. (d) Figure 9 and Figure 10 can be placed as Figure 9A, B.

(b)  Below manuscripts are suggested to strengthen the work.

(c)  Tannin-reinforced iron substituted hydroxyapatite nanorods functionalized collagen-based composite nanofibrous coating as a cell-instructive bone-implant interface scaffold - ScienceDirect

(d)  Slide-Ring Structure-Based Double-Network Hydrogel with Enhanced Stretchability and Toughness for 3D-Bio-Printing and Its Potential Application as Artificial Small-Diameter Blood Vessels | ACS Applied Bio Materials

Reviewer 3 Report

Li et al., prepared the Hyflon AD/polypropylene composite membrane for artificial lung. They characterized the membrane with various analyses. I think it is so interesting for readers and can be published after revision.

1. Abstract must be enriched with more data.

2. Novelty of the current work is not clear.

3. Would you please mention "Cost effective" of the prepared membrane in this project relative to commercial material in the market.

4. Can  use of LDPE and PTFE polymers instead of PP in the preparation of membrane, because of the LDPE and PTFE are hydrophobic polymers

5.One of the important test for using of materials in the biomedical application is MTT assay. The authors should have studied the cytotoxicity of the final sample by MTT assay.

6. Please use of the "Blend" term instead of "Composite"

Round 2

Reviewer 1 Report

Accept

Reviewer 2 Report

All the comments have been answered and the manuscript looks suitable for publishing in Membrane journal.

Reviewer 3 Report

I recommend the accept the manuscript in the current form.